



# Quantifying interdecadal changes in large-scale patterns of surface air temperature variability

Dario A Zappalà[1], Marcelo Barreiro[2], and Cristina Masoller[1]

[1]Departament de Física, Universitat Politècnica de Catalunya, Terrassa, Barcelona, Spain
[2]Departamento de Ciencias de la Atmosfera, Universidad de la Republica, Montevideo, Uruguay

*Correspondence to:* Cristina Masoller (cristina.masoller@upc.edu)

**Abstract.** We study daily surface air temperature (SAT) reanalysis in a grid over the Earth surface and identify and quantify changes in SAT variability patterns during the period 1979–2016. By analyzing Hilbert amplitude and frequency we identify the regions where relative variations are most pronounced (larger than $\pm 50\%$ for the amplitude and $\pm 100\%$ for the frequency). Amplitude variations are interpreted as due to changes in precipitation or ice melting; frequency variations, to a northward

5  shift of the inter-tropical convergence zone (ITCZ) and a widening of the rainfall band in the western Pacific Ocean. The ITCZ is the ascending branch of the Hadley cell and thus, by affecting the tropical atmospheric circulation, ITCZ migration has far reaching climatic consequences. As the methodology proposed here can be applied to many other geophysical time series, our work will stimulate new research that will advance the understanding of climate change impacts.

## 1 Introduction

The unprecedented intensification of weather extremes is motivating intensive research aimed at understanding long-term climatic variations (Barreiro et al., 2008; Coumou and Rahmstorf, 2012; England et al., 2014; Cai et al., 2014; Turco et al., 2015; Martin-Gomez and Barreiro, 2017) that can have profound socio-economic impacts and trigger complex ecological adaptation mechanisms (Lejeune et al., 2002; Beaumont et al., 2011; Gottfried et al., 2012; Bordeu et al., 2016).

15  Quantifying interdecadal variations in surface air temperature (SAT) statistical properties is a challenging problem because of the presence of long range correlations and multiple time scales (Franzke, 2012; Massah and Kantz, 2016). Data analysis tools commonly used for the study of complex systems are successfully being used for climate data analysis (Huang and Wu, 2008; Ghil et al., 2011; Palus, 2014; Tsonis and Swanson, 2008; Donges et al., 2009; Fountalis et al., 2015; Tantet and Dijkstra, 2014). In particular, networks constructed from SAT temporal inter-dependencies in different regions have yield new

20  insights into SAT variability (Barreiro et al., 2011; Palus et al., 2011; Donges et al., 2015; Stolbova et al., 2016; Tirabassi and Masoller, 2016). Local changes in SAT seasonal cycle have also been investigated and a trend to reduced cycle amplitude has been detected in many regions  (Stine et al., 2009; Qian et al., 2011; Dwyer et al., 2012; Stine and Huybers, 2012; Duan




et al., 2017; Chambers et al., 2013; Wang and Dillon, 2014); however, variations in the *fast* SAT dynamics (e.g. in daily SAT variability) remain poorly understood, and slow changes such as those observed over several decades in Fig. 1, have not yet been investigated at a global scale.

To fill this gap we perform a systematic investigation of how large-scale patterns of daily SAT variability have changed in
the last decades. We apply Hilbert analysis (Huang et al., 1998) to daily SAT reanalysis covering the Earth surface in the period 1979–2016. Our goal is to identify the most sensitive regions ("hotspots") where interdecadal changes of daily SAT variability are most pronounced.

Hilbert analysis is a powerful tool to study the output signals of complex systems (Pikovsky et al., 2001; Huang and Wu, 2008; Huang et al., 2009; Massei and Fournier, 2012; Palus, 2014; Sun, 2015; Janga Reddy and Adarsh, 2016; Schwabedar
and Kantz, 2016). It provides, for a real oscillatory signal, $x(t)$, an instantaneous amplitude, $a(t)$, and frequency, $\omega(t)$. If the signal does not have a sufficiently narrow frequency band, $a(t)$ and $\omega(t)$ may not be well defined. The usual solution is based on band-pass filtering to isolate a narrow frequency band; however, the Hilbert transform directly applied to the signal can still yield useful information.

We have recently applied the Hilbert transform to unfiltered daily SAT reanalysis (Zappalà et al., 2016). We have shown
that the maps of the time-averaged Hilbert frequency, $\langle\omega\rangle$, and of the standard deviation, $\sigma_\omega$, revealed well-defined large-scale structures which were consistent with known dynamical processes.

Here we use $a(t)$ and $\omega(t)$ to quantify interdecadal SAT variations. Our hypothesis is that changes in $a(t)$ and $\omega(t)$ can yield information about variations in SAT fast dynamics. Specifically, we are interested in addressing the following questions: Which properties of $a(t)$ and $\omega(t)$ display relevant variations? Where are the regions where these variations are more pronounced?
Which processes can be responsible of these variations? Can these variations be used as a quantitative measure of regional climate change?

## 2   Data

In the main text we present results from ERA-Interim daily SAT reanalysis (Dee et al., 2011) that covers the period from January 1979 to June 2016 with a spatial resolution of 2.5 degrees, both in latitude and in longitude. Thus, there are $N = 73 \times 144 =$
10512 geographical sites and in each site the SAT time series has $T = 13696$ days. In the *Supplementary Information* we compare ERA-Interim with NCEP-DOE Reanalysis 2 that is an improved version of the NCEP Reanalysis I model (Kistler et al., 2001) that covers a longer time interval and has $94 \times 192 = 18048$ geographical sites. In order to perform a precise comparison between the results of the two datasets, in the NCEP-DOE Reanalysis 2 we consider the same time interval as the ERA-Interim dataset.

To indicate the raw SAT time series we use the notation $r_j(t)$, where $j \in [1, N]$ represents the geographical site and $t \in [1, T]$ represents the day. From each $r_j(t)$ we calculated the *climatology* (or seasonal cycle), $c_j(t)$, and the *anomaly* time series, $z_j(t)$. We detrended and normalised (to zero mean and unit variance) $r_j(t)$, obtaining $x_j(t)$.



## 3 Methods

### 3.1 Hilbert analysis

The Hilbert transform was applied to $x_j(t)$ (Zappalà et al., 2016), obtaining the complementary oscillation $y_j(t)$. From $x_j(t)$ and $y_j(t)$, $a_j(t)$ and $\varphi_j(t)$ were calculated as: $a_j(t) = \sqrt{[x_j(t)]^2 + [y_j(t)]^2}$ and $\varphi_j(t) = \arctan(y_j(t)/x_j(t))$. By unwrapping the phase and calculating the derivative we obtained $\omega_j(t)$. Since Hilbert algorithm (Bilato et al., 2014) gives errors in the extremes, we disregarded the initial 5% and the final 5% of every series. In this way, we have time series of length $T = 12328$ with the property that $a_j(t) \cos \varphi_j(t)$ *exactly reconstructs* the normalised series $x_j(t)$ (the cross-correlation coefficient between $x_j(t)$ and $a_j(t) \cos \varphi_j(t)$ being equal to one).

### 3.2 Interdecadal Variations

To quantify interdecadal variations we calculated the difference, $\Delta a = \langle a \rangle_1 - \langle a \rangle_f$, between the average value of amplitude during the first 10 years of the time series (January 1979 to December 1988), $\langle a \rangle_f$, and the last ten years (July 2007 to June 2016), $\langle a \rangle_1$, and calculated the relative change, $\Delta a / \langle a \rangle$, where $\langle a \rangle$ is the amplitude averaged over all the years. Analogously, we calculated the interdecadal relative change of amplitude variance, $\Delta \sigma_a^2 / \sigma_a^2$, average frequency, $\Delta \omega / \langle \omega \rangle$, and frequency variance, $\Delta \sigma_\omega^2 / \sigma_\omega^2$.

A similar analysis was performed to detect interdecadal changes directly from SAT data, by computing the amplitude of the climatology and the variance of the anomaly. Specifically, the amplitude of the climatology was calculated as: $a_j^{(\text{clim})}(I) = \max[c_j^I(t)] - \min[c_j^I(t)]$, where $c_j^I(t)$ is the climatology series calculated only in the time interval $I$. We remark that the climatology amplitude $a_j^{(\text{clim})}(I)$ is a scalar number that depends on the choice of the time interval $I$. We calculated the climatology amplitude in the first and last decade, as well as in the whole series. As before, we used these values to calculate the relative change $\Delta a^{(\text{clim})} / a^{(\text{clim})}$. Also, the variance of the anomaly time series $z_j(t)$ was calculated and then used to find the relative interdecadal change, $\Delta \sigma_z^2 / \sigma_z^2$.

With the goal of relating changes in Hilbert frequency with changes in statistical properties of SAT time series, an analysis of the number of zero-crossings was performed: for each $x_j(t)$ we counted the number of crossings through the mean value, $x = 0$. As with other quantities, we then calculated the relative change.

### 3.3 Significance Analysis

A statistical significance analysis was performed by surrogating Hilbert series. For each amplitude time series (i.e., in each grid point) 100 shuffle surrogates were generated and for each surrogate the relative interdecadal change, $\Delta a^s / \langle a^s \rangle$, was calculated. Then, the average over the 100 surrogates, $\langle \Delta a^s / \langle a^s \rangle \rangle_s$, and its standard deviation, $\sigma_s$ were used to define the significance threshold: the relative change computed from the original data was considered significant if it was higher than $\langle \Delta a^s / \langle a^s \rangle \rangle_s + 2\sigma_s$, or lower than $\langle \Delta a^s / \langle a^s \rangle \rangle_s - 2\sigma_s$. In the color maps, regions where variations are not significant are displayed in white. The same test was applied to frequency variations and the other quantities, except for the climatology for





which a surrogate test is not applicable. In the *Supplementary Information* various thresholds are considered and it is shown that the results are robust with respect to the threshold.

## 4 Results

We analyze the maps of $\langle \omega \rangle$, $\langle a \rangle$, $\sigma_\omega^2$ and $\sigma_a^2$, in the first ten years and in the last ten years of the period covered by the
reanalysis, as well as the interdecadal variations.

### 4.1 Analysis of Amplitude Variations

Figures 2(a) and 2(b) display $\langle a \rangle$ in the first and in the last ten years, respectively, and Fig. 2(c) displays the relative difference (see *Methods* for details). In Fig. 2(c) we see an area of large increase (more than 50%) of average amplitude, located in South America (red spot marked by a triangle), and an area of large decrease (again, more than 50%), located in the Arctic (blue spot
marked by a circle). The raw SAT time series in these regions are displayed in Figure 1.

In both time series we clearly observe a change in the amplitude of the fast oscillations in the last ten years with respect to the first ten years, having a visual confirmation of the changes detected by Hilbert amplitude. The red spot in Amazonia, whose SAT series shown in Fig. 1(a) has an increasing amplitude, can be interpreted in terms of changes in precipitation. In particular, the increase of Hilbert amplitude is linked to the decrease of precipitation and to the lengthening of the dry season (as reported
in (Gu et al., 2016; Liebmann et al., 2004; Fu et al., 2013)). This is due to the fact that, when precipitation decreases, the fraction of solar radiation that is not used for evaporation is used to heat the ground, which in turns heats the surface air. Regarding the blue spot in the Arctic region, where SAT series shown in Fig. 1(b) has a decreasing amplitude, it can be interpreted as due to the melting of sea ice. In fact, when ice is present at the surface of the sea, it acts as an insulator, preventing heat exchange between sea and air. This causes a large amplitude of SAT cycle. On the other hand, if the ice melts, the air-sea heat exchange
reduces the amplitude of the cycle. In particular, during winter the air temperature is mitigated by the sea and tends to have more moderated values.

Next, we compare the changes detected by Hilbert amplitude with those computed directly from SAT (by decomposing SAT time series into climatology and anomaly, as explained in *Methods*). Since the climatology term retains the seasonal variation, we expect its amplitude change to give similar indications as the Hilbert amplitude change. On the other hand, the anomaly
term contains all the rapid variability, so we expect its variance to give similar results as the variance of Hilbert amplitude.

Figures 2(c) and 2(d), which display $\Delta a / \langle a \rangle$ and the relative change of climatology amplitude, respectively, and Figs. 3(a) and 3(b), which display $\Delta \sigma_a^2 / \sigma_a^2$ and the relative change of the anomaly variance, respectively, confirm these expectations.

The good qualitative agreement seen in the spatial structures in these maps confirms that Hilbert analysis directly applied to unfiltered SAT indeed gives a physically meaningful instantaneous amplitude, with average and variance values that are
consistent with those computed from SAT.

In Figs. 3(a) and 3(b), however, there is a difference in the eastern Pacific Ocean, in the area marked with a circle. In particular, in Fig. 3(b) there is an area with large decrease of variance (deep blue, around -100%), while in (a) the decrease is



less pronounced (light blue, around -65%) and extended over a smaller area. In addition, in Fig. 3(a) there is an orange-red area that indicates a moderate increase of variance (around 45%), while in (b) such area is absent. The reasons underlying these differences will be discussed later.

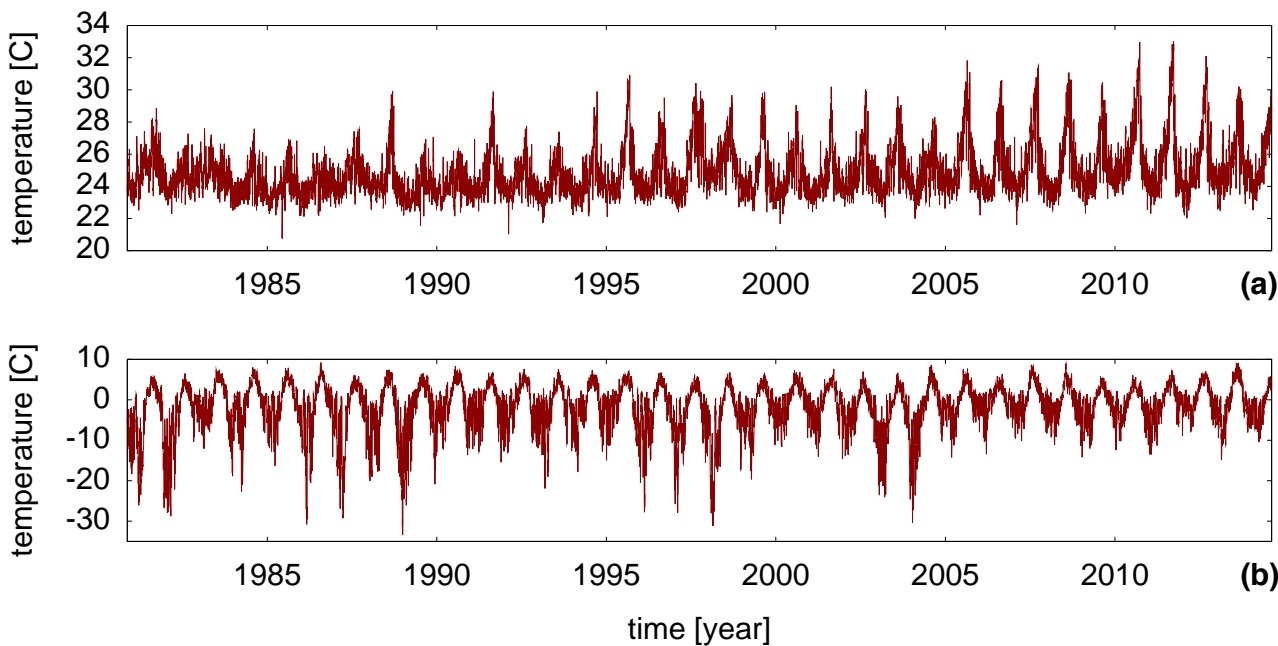

**Figure 1.** Surface air temperature in two regions where a clear change in the oscillation amplitude in the last ten years, with respect to the first ten years, is observed. (**a**) Site of coordinates (7.5 S, 307.5 E), marked with a triangle in Fig. 2(c). (**b**) Site of coordinates (75 N, 40 E), marked with a circle in Fig. 2(c).

### 4.2 Analysis of Frequency Variations

Figure 4(a) displays the average frequency $\langle\omega\rangle$ in the first ten years, Fig. 4(b) in the last ten years, and Fig. 4(c) displays the relative change, $\Delta\omega/\langle\omega\rangle$. In Fig. 4(c) we note that in the eastern Pacific Ocean there are two small areas, enclosed by the circle, of intense increase (red) and decrease (blue) of frequency. They both represent frequency changes whose absolute values are larger than 100% and correspond to the same region where differences were detected in Fig. 3.

These two areas of opposite signs suggest that, between the initial and the final decade, there is a shift of the inter-tropical convergence zone (ITCZ) toward the north. The ITCZ involves strong convective activity, which causes rapid fluctuations of SAT, thus giving high values of instantaneous frequency, as shown in Figs. 4(a,b). Therefore, in the relative change of frequency, in regions corresponding to the initial position of the ITCZ we see a decrease, while in regions corresponding to the present position of the ITCZ we see an increase. For the same reason, the two red areas in the western Pacific Ocean (indicated by two squares) suggest an expansion of the tropical convective regions. This interpretation is in agreement with recent works



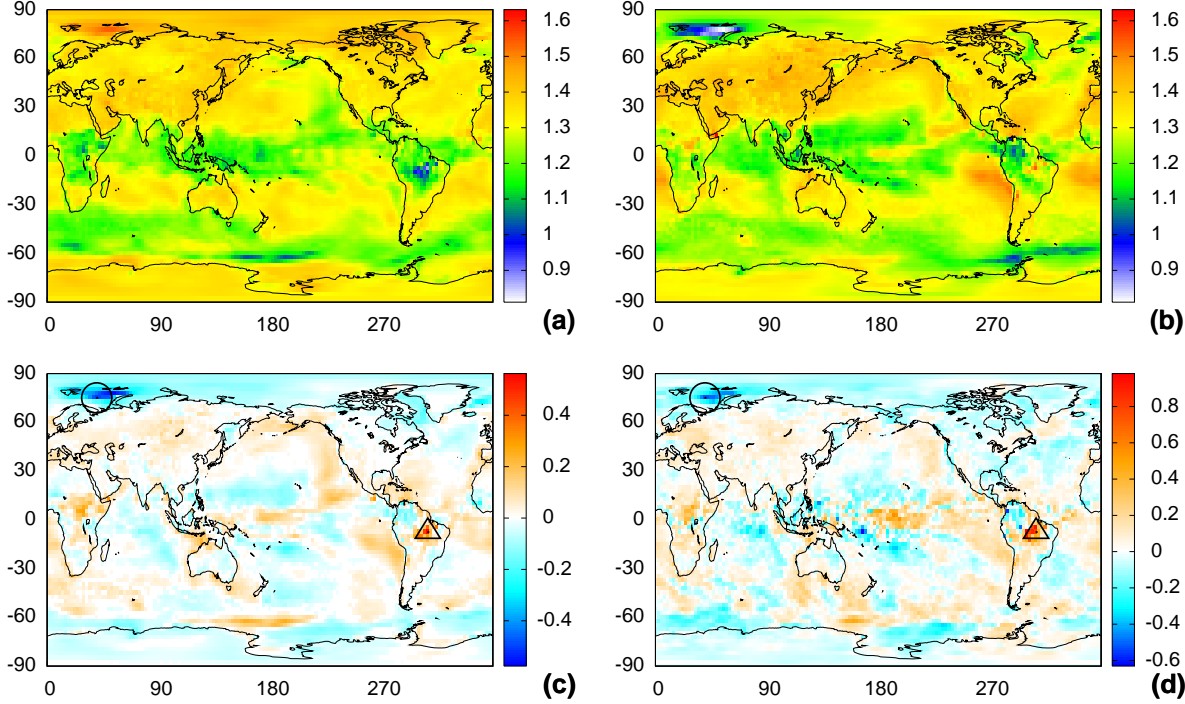

**Figure 2.** Relative change of time-averaged Hilbert amplitude. (**a**) Amplitude averaged over the first ten years (January 1979 to December 1988). (**b**) Amplitude averaged over the last ten years (July 2007 to June 2016). (**c**) Relative change of Hilbert amplitude, $\Delta a/\langle a\rangle$. (**d**) Relative change of amplitude of the seasonal cycle, computed from the amplitude of the climatology, $\Delta a^{(\mathrm{clim})}/a^{(\mathrm{clim})}$. A good qualitative agreement is seen in the spatial structures in panels (c) and (d). Importantly, the structures uncovered by Hilbert amplitude are well defined, in comparison with those uncovered by the analysis of the climatology amplitude, which look noisier.

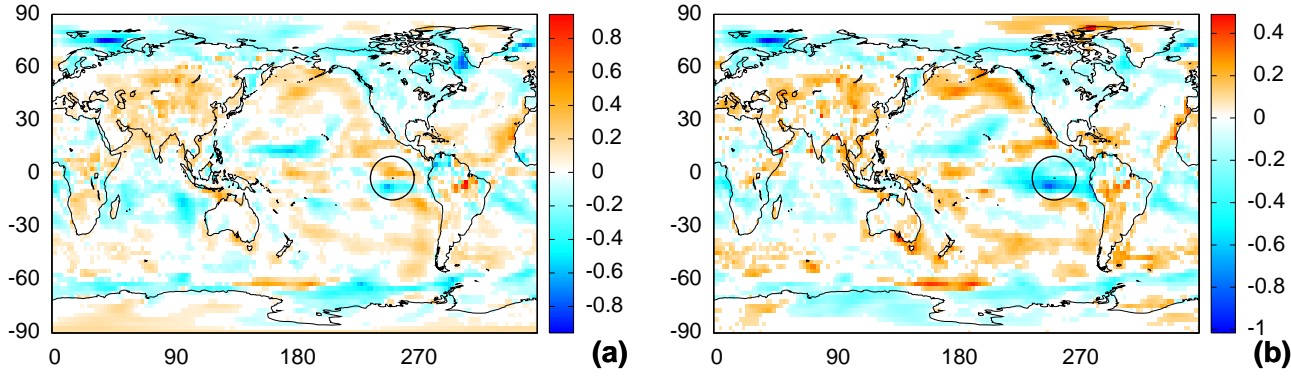

**Figure 3.** Relative change of amplitude fluctuations computed from the variance of (**a**) Hilbert amplitude, $\Delta\sigma_a^2/\sigma_a^2$; (**b**) the anomaly time series, $\Delta\sigma_z^2/\sigma_z^2$.

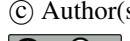



that have identified a northward shift of ITCZ (Talento and Barreiro, 2016; Yoshimori and Broccoli, 2008; Kang et al., 2009; Frierson and Hwang, 2012; Schneider et al., 2014). Regarding the red areas in the north Atlantic, in the north Pacific and in the south Pacific, they are consistent with an increase in the occurrence of fronts which cause large daily fluctuations of temperature and thus an increase of Hilbert frequency.

To gain insight into the physical meaning of the changes that are captured by Hilbert frequency, we use an alternative approach to estimate frequency variations: we define as "events" the zero crossings of SAT time series (Pikovsky et al., 2001) (detrended and normalized to zero-mean as described in *Methods*). Then, we count the number of events in the first ten years, in the last ten years, and calculate the relative variation.

Figure 4(d) displays the map of relative change of zero-crossings. We see that there is a qualitative good agreement with the
spatial structures seen in Fig. 4(c), thus providing a physical interpretation for the observed variation of Hilbert frequency: the areas where the frequency increases (decreases) correspond to areas where the number of zero-crossings increases (decreases). We note that the relative variations in Hilbert frequency are more pronounced than those in the number of crossings. We note that this specifically holds in the regions where frequency variations are interpreted in terms of ITCZ migration.

Figures 5(a) and 5(b) display SAT time series in the dipole region indicated with the circle in Fig. 4(c), and also indicate
(in red) the zero-crossings. We can understand the difference that was detected in this region between the variance of Hilbert amplitude (Fig. 3a) and the variance of anomaly (Fig. 3b). This difference is explained in the following terms: in the first decade the seasonal cycle is more irregular than in the last decade, probably a consequence of an El Niño event in 1982–1983. The anomaly series contains these slow fluctuations as well as the rapid ones, and thus its variance is affected by both effects. In contrast, the Hilbert amplitude is less affected by the slow fluctuations as its variance captures mainly the rapid fluctuations
of SAT.

To demonstrate the robustness of our findings, in the *Supplementary Information* we compare the results obtained from ERA-Interim with those obtained from another reanalysis dataset, NCEP-DOE. We find a good qualitative agreement in the spatial structures in the maps of $\langle\omega\rangle$, $\langle a\rangle$, $\sigma_\omega^2$ and $\sigma_a^2$, but we discuss also some relevant differences. In addition, to further understand the relationship between statistical properties of SAT and those of Hilbert amplitude and frequency, in the *Supplementary*
*Information* we apply Hilbert analysis to synthetic data generated by an autoregressive AR(1) process. We chose an AR(1) process because it is commonly used in the literature to model climate data. We find that, when increasing the noise intensity in the synthetic series, the Hilbert amplitude decreases while the frequency increases and show that this trend is also observed in real SAT time series.

## 5   Conclusions

We have used Hilbert analysis to uncover interdecadal changes in SAT daily time series. Large variations of Hilbert amplitude (more than 50%) in the Arctic and in Amazonia were interpreted respectively as due to ice melting and precipitation decrease. The analysis of Hilbert frequency also uncovered areas of large changes. In particular, two areas of opposite changes in eastern Pacific Ocean and two areas of increase in western Pacific Ocean suggest a shift towards north and a widening of the ITCZ.





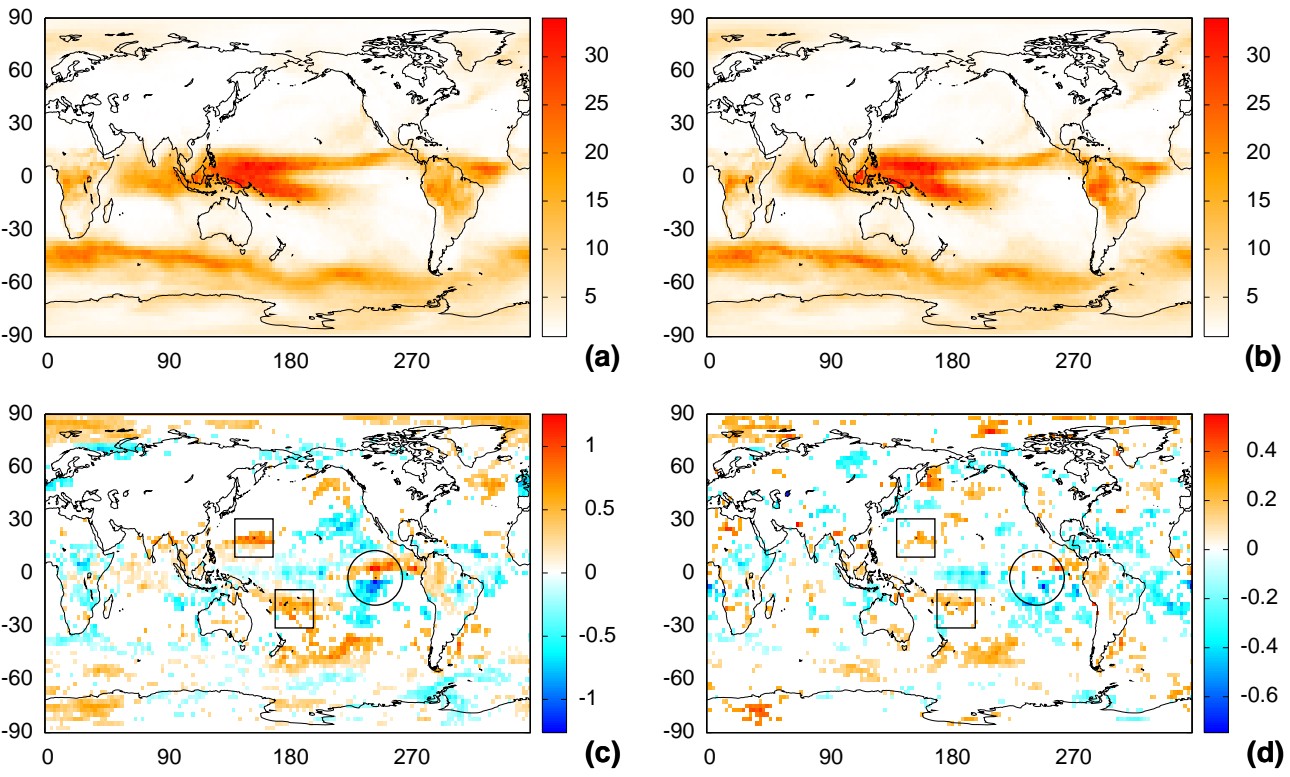

**Figure 4.** Relative change of time-averaged Hilbert frequency (in units of oscillations/year). (**a**) Average in the first ten years (1979–1988). (**b**) Average in the last ten years (2007–2016). (**c**) Relative change of Hilbert frequency, $\Delta\omega/\langle\omega\rangle$. (**d**) Relative change of the number of zero-crossings of the normalized SAT time series. In (a) and (b) the color scale is adjusted to represent in white the regions where the average frequency is one oscillation per year. In (c) and (d), a good qualitative agreement of spatial structures is seen; however, we note that Hilbert frequency detects stronger variations than those measured by the number of zero-crossings.

While there is evidence that ITCZ has moved north-south in the past, to the best of our knowledge our work is the first one to confirm this migration in the XX century. Our findings have important implications because, as the ITCZ is the ascending branch of the Hadley cell, its migration affects both the Earth's radiative balance and the release of latent heat that drives the tropical atmospheric circulation. Taken together, these effects have not only local but far reaching climatic consequences.

5   Additional analysis provided in the *Supplementary Information* confirms the robustness of these observations.

As the methodology used here can be applied to many other climatological time series that exhibit well defined oscillatory behavior, we believe that our work will stimulate new research to identify and quantify the impacts of climate change, directly from observed data.

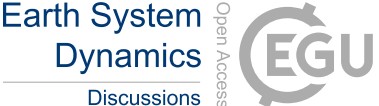



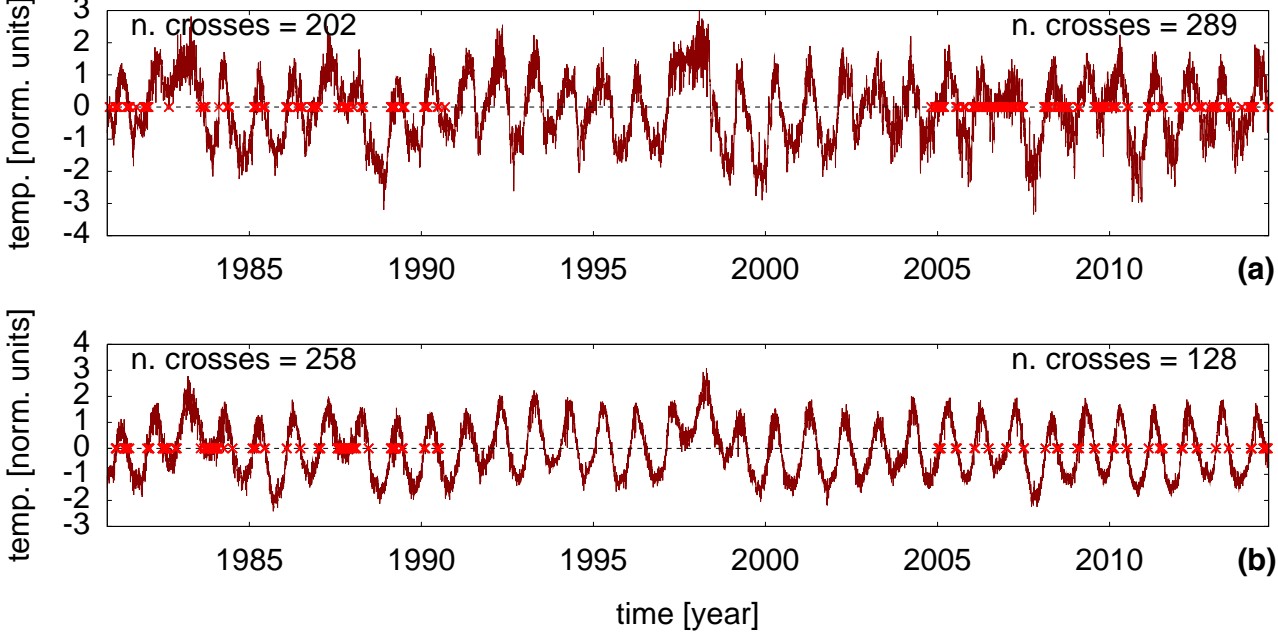

**Figure 5.** Normalised SAT time series and number of zero-crossings in the regions indicated with a circle in Fig. 4(c). In the red region (2.5 N, 245 E), panel (**a**), the number of zero-crossings increases in the last ten years with respect to the first ten years (289/202 respectively), while in the blue region (7.5 S, 250 E), panel (**b**), it decreases (128/258 in the last/first ten years).

*Code and data availability.* The Hilbert algorithm used is available in Bilato et al. (2014); the data sets used are ERA-Interim Reanalysis, provided by the European Centre For Medium-Range Weather Forecasts (ECMWF), Reading, UK, from their website: https://www.ecmwf.int and NCEP-DOE Reanalysis 2, provided by NOAA Boulder, Colorado, USA, from their website: http://www.esrl.noaa.gov/psd/.

*Competing interests.* The authors declare no conflict of interest.

5    *Acknowledgements.* This work was supported in part by the LINC project (EU-FP7-289447), ICREA ACADEMIA (Generalitat de Catalunya, Spain), and by Spanish MINECO/FEDER (FIS2015-66503-C3-2-P). D. A. Zappala also thanks European Social Fund and the Generalitat de Catalunya for the FI-AGAUR scholarship.




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
