# Peer review of "Quantifying changes in spatial patterns of surface air temperature dynamics over several decades"

_Earth System Dynamics, 2017_

## Short Comment (SC1) · 21 Sep 2017

We would like to include these videos to provide additional information that demonstrates the validity of Hilbert analysis:

https://www.dropbox.com/sh/87txchq7aoibjst/AAC-4fv2lNYt5Ol1RvTou17oa?dl=0

The three videos show in a map the time-averaged evolution of cos(phase) along a typical year (i.e. averaging cos(phase) over all the years), a El Niño year (averaged selecting only periods of El Niño) and a La Niña year (averaged selecting only periods of La Niña).

[Figure]

2017.

---

## Referee Comment (RC1) · Anonymous Referee #1 · 19 Oct 2017

Comments on the manuscript entitled "Quantifying interdecadal changes in large-scale patterns of surface air temperature variability" by Zappalà et al.

The authors try to quantify interdecadal changes in large-scale patterns of surface air temperature (SAT) variability using the Hilbert transform. The authors obtain instantaneous amplitude and frequency of a time series by applying Hilbert transform to the daily SAT data. Afterwards the instantaneous amplitude and frequency are averaged and compared to quantify the differences in SAT amplitude and frequency between two decades (1979-1988 VS 2007-2016). This methodology has some fatal flaws (please see the following major comments). I did not find the merit of the proposed methodology compared to other commonly used methods. The study is also contradicting the title of the manuscript. Given these reasons, the manuscript does not meet the stan-

dard of an international journal in its current form. I have to vote for rejection of the manuscript at this time.

Major comments: The methodology proposed in the manuscript has some fatal flaws: (1) The period of decadal- or interdecadal-scale climate variability is not necessarily equal to ten years in nature. It may vary in a very large range. The phase may be different at different geographic locations. For example, Pacific Decadal Oscillation (PDO) and Atlantic Multidecadal Oscillation (AMO) are very significant decadal and multidecadal variabilities in climate system. They have very different periods and geographic locations. Therefore, it is inappropriate that the authors compare the data between two fixed time periods, i.e. 1979-1988 and 2007-2016, when quantifying the interdecadal changes of SAT. In terms of the extraction of climate variability, a successful signal processing method, e.g. FFT, wavelet transform, should be able to automatically detect the amplitude, period, and phase of a time series at various time scales. The author proposed method includes signals at all time scales and does not filter out the interdecadal variability. In this respect, I do not see the merit of the method proposed in the manuscript. I noticed that the authors cited some literatures with regard to Hilbert-Huang transform (HHT). HHT, consisting of empirical mode decomposition (EMD) and Hilbert spectral analysis, can provide a time-frequency-energy description of a time series, which has been used extensively in geophysical research. Why the authors only use Hilbert transform here? In terms of the detection of interdecadal climate variability, are there any merits of the proposed methodology compared to the HHT? I personally do not think the authors present an effective method in identifying interdecadal climate variabilities. If the authors think they did, they have to clearly elucidate the merits of their approach beyond other data analysis methods.

The introduction was not well written. There are many general sentences followed by a number of literature citations without going into details. For example, in P1 L16-19, "Data analysis tools commonly used for the study of complex system are successfully being used for climate data analysis (Huang and Wu, 2008; Ghil etal., 2011; Palus,

2014; Tsonis and Swanson, 2008; Donges et al., 2009; Fountailis et al., 2015; Tantet and Dijkstra, 2014)". What data analysis methods or tools do the authors refer to here? What is the strength and shortcoming of various methods? These general sentences barely provided any useful information to readers.

The significance test proposed by the authors lacks mathematical basis. What is the significance level? The determination of significance threshold seems arbitrary. Why $2\sigma$s is chosen?

The title of manuscript includes two key words, "interdecadal changes" and "larges-scale" patterns". However, as I previously explained, it is inappropriate to measure the interdecadal changes in SAT with the difference between two fixed periods. The interdecadal changes of SAT presented in the study are misleading due to the methodology. On the other hand, the manuscript focuses on some specific spots, e.g. the spots located in Arctic and Amazonia. This is not large-scale pattern of SAT. Therefore, the research contradicts the title of the manuscript. The authors could consider focusing on the large-scale patterns of SAT, e.g. the PDO- or AMO-related SST pattern. However, I do not see very clear PDO- or AMO-like SAT pattern from the figures, which is probably due to the inappropriate approach employed, i.e. using the difference between two fixed periods. For example, no clear difference in PDO-like SAT pattern can be found if both periods (1979-1988 and 2007-2016) were in the positive phase of PDO. Under such circumstance, the conclusion of no PDO-like decadal variability would be clearly wrong.

The explanations on the reasons of interdecadal changes in SAT daily time series are vague and hand waving. I'm not saying the explanations are wrong but lack of in-depth analysis and evidence. Some explanations do not even match the results shown in the figures. For example, figure 1 shows a clear increase in the amplitude of SAT time series. The authors argue that the increase in the amplitude of SAT series is linked to the decrease in precipitation. The decrease in precipitation in turn leads to an increase in SAT due to changes in energy partition between latent heat and sensible heat (P4,

L12-16). Assume this is correct, but it only explains the increase in SAT rather than the increase in the "amplitude" of SAT.

The conclusion section did not well summarize the main findings. In my point of view, the key point of the manuscript is the method proposed to identify climate variability using Hilbert transform. However the short conclusion section barely summarizes the method.

What does the "confirm this migration in the XX century" mean in P8, L2?

––––––––––––––––––––––––––––

---

## Author Comment (AC1) · 23 Oct 2017

Anonymous Referee #1 says a) This methodology has some fatal flaws (please see the following major comments). b) I did not find the merit of the proposed methodology compared to other commonly used methods. c) The study is also contradicting the title of the manuscript.

As in the discussion stage authors are "invited to take an active role in the debate by posting author comments as a response to referee comments and short comments of the scientific community as soon as possible" we aim to contribute to the discussion by addressing each of these points.

Regarding point (a) the Referee says "The methodology proposed in the manuscript

has some fatal flaws: (1) The period of decadal- or interdecadal-scale climate variability is not necessarily equal to ten years in nature. It may vary in a very large range. The phase may be different at different geographic locations. For example, Pacific Decadal Oscillation (PDO) and Atlantic Multidecadal Oscillation (AMO) are very significant decadal and multidecadal variabilities in climate system. They have very different periods and geographic locations."

Of course, we agree with the referee in his/her comment that climate variability has different periods, in different geographic locations. However, this does not mean that the "methodology proposed in the manuscript has some fatal flaws". In our manuscript, we don't refer to (or claim to analyze) "The period of decadal- or interdecadal-scale climate variability" as clearly such a period cannot be analyzed with SAT time-series that cover only 36 years (as in our manuscript). Regarding the phase of PDO and AMO, we remark that we also do not refer to them in our manuscript because, as we just said, we cannot analyze such phenomena (with decadal or longer time-scale) with only 36 year of data.

Regarding the methodology used, Hilbert analysis, we remark that is well established, and can be applied to signals that have a well-defined periodicity (in our case, SAT data which has a characteristic period of one year).

The reviewer says "Therefore, it is inappropriate that the authors compare the data between two fixed time periods, i.e. 1979-1988 and 2007-2016, when quantifying the interdecadal changes of SAT".

It seems to us that the key word "interdecadal" must have been misleading to the reviewer. In our manuscript, we compare variations in average magnitudes (and their standard deviations) between the first ten years and the last ten years of the dataset. We cannot think of a single argument by which this could be "inappropriate" or have "some fatal flaws". Moreover, in a revised version of the manuscript we can add several references where a similar comparison (between two ten-year periods) is performed.

We could also compare the first and last five years of the dataset (i.e. 1979-1983 and 2002-2016), or we could compare the first half and the last half of the data set (i.e. 1979-1999 and 2000-2016). There is nothing intrinsically wrong in doing that, and moreover, we show in our manuscript that some interesting and relevant conclusions can be extracted from such analysis.

The reviewer says "In terms of the extraction of climate variability, a successful signal processing method, e.g. FFT, wavelet transform, should be able to automatically detect the amplitude, period, and phase of a time series at various time scales".

We fully agree with this comment but that is not scope of our work, as we aim to detect instantaneous amplitude, frequency and phase. To the best of our knowledge, only the Hilbert transform can provide, for each data value, $x(t)$, in a time-series, an instantaneous amplitude, $a(t)$, frequency, $f(t)$ and phase, $fi(t)$.

The reviewer says "The author proposed method includes signals at all time scales and does not filter out the interdecadal variability. In this respect, I do not see the merit of the method proposed in the manuscript."

A main point in our manuscript is that we demonstrate that meaningful information can be extracted from the raw data, without making any a priori assumption, i.e., without the need to filter the data (we preprocess the data just by de-trending and normalizing).

The reviewer says "I noticed that the authors cited some literatures with regard to Hilbert-Huang transform (HHT). HHT, consisting of empirical mode decomposition (EMD) and Hilbert spectral analysis, can provide a time-frequency-energy description of a time series, which has been used extensively in geophysical research. Why the authors only use Hilbert transform here?"

Our aim is to show that a simple approach can indeed yield meaningful information. To keep the algorithm simple we have not used the HHT. However, we performed extensive tests to verify that the amplitude and the phase extracted were appropriated

(x(t)=a(t) cos[fi(t)] for all t except during a transient first and final time-intervals that were disregarded). We also compared the results with different algorithms that implemented the Hilbert transform and different ways of computing the Hilbert phase. Moreover, in the supplementary information we compared two reanalysis and did not detect any "fatal flaws" or contradiction with well-known climate phenomena, to justify trying a more advanced method. In addition, in the supplementary information we also show that results of Hilbert analysis can be well-explained by a simple model (AR(1) process) which is commonly used as a null-hypothesis in climate studies.

Of course, this argument is valid for the analysis of SAT data that, as mentioned before, has a defined periodicity imposed by the annual solar cycle. The analysis of other climatological variables (lacking well defined periodicity) will likely need the use of more advanced tools, such as HHT.

The reviewer says "In terms of the detection of interdecadal climate variability, are there any merits of the proposed methodology compared to the HHT?"

As we explained before we limit the study to detect changes (in averaged quantities and standard deviations) between the first and last ten years of the dataset. In this sense, a main merit is that the algorithm, while being simple to implement, gives information which is consistent with information we extracted by other means (see next paragraph).

Regarding point (b): "I did not find the merit of the proposed methodology".

The reviewer says "I personally do not think the authors present an effective method in identifying interdecadal climate variabilities. If the authors think they did, they have to clearly elucidate the merits of their approach beyond other data analysis methods."

In our manuscript we compare the results of Hilbert analysis with another data analysis method (as explained in section Methods, subsection 3.2). Specifically, we compare two approaches in Figs. 2(c) and 2(d), in Fig. 3(a) and 3(b) and in Fig. 4(c) and 4(d). In all cases we have found consistent results. Regarding the merits of our approach, we

note that Hilbert analysis detects stronger variations, and/or more clear (better defined) spatial patterns: the uncovered regions where variations are more pronounced are clearer –i.e., less noisy- with the Hilbert approach.

The reviewer says "The introduction was not well written. There are many general sentences followed by a number of literature citations without going into details. For example, in P1 L16-19, "Data analysis tools commonly used for the study of complex system are successfully being used for climate data analysis (Huang and Wu, 2008; Ghil etal., 2011; Palus, 2014; Tsonis and Swanson, 2008; Donges et al., 2009; Fountailis et al., 2015; Tantet and Dijkstra, 2014)". What data analysis methods or tools do the authors refer to here? What is the strength and shortcoming of various methods? These general sentences barely provided any useful information to readers."

The goal of this sentence was to provide the reader with a non-exhaustive list of relevant references; however, we agree with the reviewer that the information provided is barely useful and we will be happy to expand and improve the introduction in a revised version of the manuscript.

The reviewer says "The significance test proposed by the authors lacks mathematical basis. What is the significance level? The determination of significance threshold seems arbitrary. Why 2sigma s is chosen?"

It seems to us that the reviewer did not notice that this manuscript is accompanied by supplementary information where we show how the significance threshold modifies the obtained maps (Fig. 2 of the SI presents results with no threshold, 2sigma and 4 sigma). We will be happy to move this information to the main manuscript in a revised version. We will also be happy to expand this section to further discuss the appropriateness of the significance test done.

Regarding point (c): The study is also contradicting the title of the manuscript.

The reviewer says "The title of manuscript includes two key words, "interdecadal

changes" and "large-scale" patterns". However, as I previously explained, it is inappropriate to measure the interdecadal changes in SAT with the difference between two fixed periods. The interdecadal changes of SAT presented in the study are misleading due to the methodology.

In the context of our work, "interdecadal" refers to changes in the last decade with respect to the first decade of the reanalysis. While we believe that it is correct to refer to such changes as "interdecadal", for the sake of clarity we will be happy to modify the wording in the title and in the text in a revised version. We suggest the following title, which more clearly reflects the content of the manuscript: Quantifying changes in surface air temperature dynamics over several decades.

The reviewer says "On the other hand, the manuscript focuses on some specific spots, e.g. the spots located in Arctic and Amazonia. This is not large-scale pattern of SAT. Therefore, the research contradicts the title of the manuscript. The authors could consider focusing on the large-scale patterns of SAT, e.g. the PDO- or AMO-related SST pattern."

In the context of our work, "large-scale pattern" refers to spatial patterns, which are well-defined geographical regions where we uncover large changes in Hilbert magnitudes (averaged amplitude, averaged frequency, and the standard deviations). As we said before, we don't mention PDO or AMO in our manuscript. Again for the sake of clarity, we will be happy to modify the wording in the title and in the text of a revised manuscript.

However, I do not see very clear PDO- or AMO-like SAT pattern from the figures, which is probably due to the inappropriate approach employed, i.e. using the difference between two fixed periods. For example, no clear difference in PDO-like SAT pattern can be found if both periods (1979-1988 and 2007-2016) were in the positive phase of PDO. Under such circumstance, the conclusion of no PDO-like decadal variability would be clearly wrong."
As we said before, we do not study changes of (we don't refer to and we do not make any conclusion of) PDO or AMO. We are convinced that the key words "inter-decadal changes" and "large-scale patterns" lead the reviewer to a misunderstanding /misinterpretation of our work and results. For the sake of clarity we will be happy to change these key words in the title and in the text of a revised manuscript.

The reviewer says "The explanations on the reasons of interdecadal changes in SAT daily time series are vague and hand waving. I'm not saying the explanations are wrong but lack of in-depth analysis and evidence. Some explanations do not even match the results shown in the figures. For example, figure 1 shows a clear increase in the amplitude of SAT time series. The authors argue that the increase in the amplitude of SAT series is linked to the decrease in precipitation. The decrease in precipitation in turn leads to an increase in SAT due to changes in energy partition between latent heat and sensible heat (P4, L12-16). Assume this is correct, but it only explains the increase in SAT rather than the increase in the "amplitude" of SAT."

Our work is aimed at i) proposing a new method to study climatic variations as those shown in Fig. 1, and ii) using this method to detect the geographical regions where such variations are more pronounced. To confirm the robustness of our results in the supplementary information we compare two reanalysis dataset. We see that the results are consistent; however, also some differences are detected and discussed. While we would like to provide more "in-deep" explanations of the changes uncovered, unfortunately this is not always possible and that is why the explanations might seem "vague and hand waving". We can of course try to improve the interpretation of our findings, however, we remark that the main goals of our work are to propose a new methodology and to identify the regions where the changes are more pronounced. We hope that our results will motivate further investigations to understand the reasons that underlie the uncovered large variations.

We disagree with "Some explanations do not even match the results shown in the figures." In particular, in the example mentioned by the reviewer, both time series

confirm the changes detected by Hilbert amplitude, and the explanation provided match the increase in the amplitude of SAT annual oscillation: because precipitation in the Amazonas has annual periodicity, the decrease of the amplitude of the precipitation cycle is likely to produce an increase in the amplitude of SAT cycle.

The reviewer says "The conclusion section did not well summarize the main findings. In my point of view, the key point of the manuscript is the method proposed to identify climate variability using Hilbert transform. However the short conclusion section barely summarizes the method."

We agree with the reviewer that a key point of the manuscript is the method proposed (to identify changes in SAT dynamics that occur over several decades), and in a revised version we will be happy to further stress this point in the conclusions.

The reviewer says "What does the "confirm this migration in the XX century" mean in P8, L2?"

In our manuscript some specific changes detected in Hilbert frequency are interpreted as due to a northward migration and enlargement of the inter-tropical convergence zone (P5, paragraph that starts in L9), a discussion that the reviewer does not seem to have noticed. We agree however that this sentence in the conclusion is unclear and we will be happy to change the wording in a revised manuscript.

Please also note the supplement to this comment:
https://www.earth-syst-dynam-discuss.net/esd-2017-79/esd-2017-79-AC1-supplement.pdf

---

## Referee Comment (RC2) · Anonymous Referee #2 · 25 Oct 2017

The authors address a novel problem relevant to the study of climate variability and change, that of changes in the short term variability in Surface Air Temperature (SAT) over the satellite era (1979-2016) from a global reanalysis. To this end they use an advanced signal processing method, Hilbert analysis.

I would like to praise the authors for having complemented the data analysis with appropriate, too often lacking, significance tests. Moreover, the authors went further than obtaining numerical results with a novel mathematical methods by giving coherent physical interpretations of the results and reaching novel conclusions regarding climate variability. They revealed the northward displacement of the intertropical convergence zone during the XXth century, which is an important result in itself and shows that the method has potential to gain novel insights on climate variability and change.

[Figure]

I suggest the following as minor revisions:

- The Hilbert transform should be better motivated. What can be achieved? Why does it give an amplitude and a phase? What are its limitations? A brief overview of how the Hilbert transform works would also be helpful to potential readers not acquainted to this method. I understand that the authors may not want to lengthen the article, but the latter could be given as first section of the supplementary material. Something as short and clear as Pikowsky et al., 2002, Appendix A.2.1, would greatly benefit the reader.

- Regarding the interpretation of the blue and red spots in Fig. 2, please discuss the quality of the reanalysis in these regions. In particular, the blue spot in the Arctic is in a region for which there is little constraints from satellites on the re-analysis.

Technical comments:

- p.2, l.32: what is meant by "detrended"? The climatology is kept, while the long term trend is removed? How is this done?

- p.3, l.4: what is meant by "unwrapping the phase"

- p.3, l.6: how the 5% where chosen? Are the results robust to this choice?

- p.3, l.7: exactly reconstructs $x_j$, but for extreme realizations, right?

- p.3, l.30: That the results are robust to the threshold is convincing enough. However, why make the threshold based on the standard deviation and not perform a fully non-parametric test by choosing a significance tolerance, say $\alpha = 0.05$ and consider as significant all values lager than $(1 - \alpha/2) * 100\%$ of surrogate realizations?

- p.4, l.11: Please, be more precise regarding what is meant by "fast oscillations".

Supporting Information:

- p.1, l.8: "In order" instead of "n order".

- Figure 2: the numbering of the panels is missing.

---

## Author Comment (AC2) · 27 Oct 2017

We thank the Anonymous Referee #2 for his/her positive comments about our work, and suggested revisions that will allow us to improve:

"The Hilbert transform should be better motivated. What can be achieved? Why does it give an amplitude and a phase? What are its limitations? A brief overview of how the Hilbert transform works would also be helpful to potential readers not acquainted to this method. I understand that the authors may not want to lengthen the article, but the latter could be given as first section of the supplementary material. Something as short and clear as Pikowsky et al., 2002, Appendix A.2.1, would greatly benefit the reader."

Authors' response: Our motivation to use of Hilbert transform (HT) is that it has been

demonstrated useful to analyse and to characterise oscillating signals of different kinds, but to the best of our knowledge, it has not yet been used to investigate changes in surface air temperature data. We have performed a first analysis in Zappala et al (2016) where we detected well defined spatial patterns in Hilbert magnitudes, and our motivation here is to determine how these patterns have changed over the last three decades. In our manuscript we have briefly commented the main limitation of HT: the difficulty in dealing with signals without a sufficiently narrow band of frequency. In a revised version of the manuscript we will be happy to extend the introduction to describe in more detail the motivation of our study, as well as the limitations of the Hilbert approach. Moreover, we agree with the reviewer that adding a section in the Supporting Information to explain the basics of HT will be very helpful to the readers.

"Regarding the interpretation of the blue and red spots in Fig. 2, please discuss the quality of the reanalysis in these regions. In particular, the blue spot in the Arctic is in a region for which there is little constraints from satellites on the re- analysis."

Authors' response: We thank the reviewer for pointing out this issue. Actually, our way to deal with possible problems with the quality of the reanalysis has been to consider two different datasets (ERA-Interim and NCEP-DOE Reanalysis 2) and compare the results. We show in Section 3 of the Supporting Information that the blue spot is in the same position in the two datasets. While this could be considered a confirmation of a genuine amplitude change, it could also be an artifact in the two reanalysis, due to the same constraints from satellites. In a revised version of the manuscript we will be happy to discuss this point.

Regarding the technical comments: p.2, l.32: what is meant by "detrended"? The climatology is kept, while the long term trend is removed? How is this done?

Authors' response: First of all, from the raw SAT time series, we calculate the linear regression, to find the long term linear shift of temperature. Then, we subtract this linear trend from the SAT series. We do this because, to analyse the oscillation of the

series, we don't want the center of oscillation to shift in time.

p.3, l.4: what is meant by "unwrapping the phase"?

Authors' response: To calculate the phase from x and y we use the arctan function. If we keep into account the sings of x and y, we get phase values in the domain [-$\pi$, $\pi$]. So, basically, the time series of phase has jumps from $\pi$ to - $\pi$. By "unwrapping the phase" we just mean to eliminate these jumps (with a standard matlab function), to obtain a series which values are not limited to the domain [-$\pi$, $\pi$].

p.3, l.6: how the 5% where chosen? Are the results robust to this choice?

Authors' response: We analysed the results of our HT algorithm over synthetic series generated by us (with known amplitude and frequency). In this way, we could compare the results given by HT with the true values of frequency and amplitude. As known by previous studies, near the two extremes of the series we found differences between HT results and the true values. We chose the value of 5% as a security threshold, because in all the tests it was sufficient to eliminate the parts of the series where HT gives significant deviations from the true amplitude and frequency values.

p.3, l.7: exactly reconstructs xj , but for extreme realizations, right?

Authors' response: We performed extensive tests and found that, in all sites, x(t) and A(t) cos$\varphi$(t) are exactly equal at each time "t", except when "t" is too close to the extremes of the time series (i.e., except a few initial and final values), which were disregarded from the analysis (the 5% explained in the previous point). In a revised version, we will be happy to describe in the detail the calculations, and include a comparison between the original SAT time series, x(t), and the series obtained from the HT, A(t) cos$\varphi$(t).

p.3, l.30: That the results are robust to the threshold is convincing enough. However, why make the threshold based on the standard deviation and not perform a fully non-parametric test by choosing a significance tolerance, say alpha = 0:05 and consider as

significant all values larger than (1 –alpha /2) * 100% of surrogate realizations?

Authors' response: We thank the reviewer for this suggestion and we will add this non-parametric test in the revised version of the manuscript.

p.4, l.11: Please, be more precise regarding what is meant by "fast oscillations".

Authors' response: Since we just meant "amplitude of oscillations", in a revised version we will remove the world "fast".

Finally, the reviewer is absolutely right about the typo and the numbering of Figure 2 of Supporting Information, and we will correct these mistakes in a revised version.

Please also note the supplement to this comment:
https://www.earth-syst-dynam-discuss.net/esd-2017-79/esd-2017-79-AC2-supplement.pdf

---

## Author Response (AR1)

Terrassa, Barcelona, Spain, December 20, 2017

Manuscript number: ESD-2017-79

Title: Quantifying changes in surface air temperature dynamics over several decades

Authors: D. Zapalla, M. Barreiro and C. Masoller

We have answered the comments of the two referees:

https://editor.copernicus.org/index.php/esd-2017-79-AC1.pdf?\_mdl=msover\_md&\_jrl=430&\_lcm=oc108lcm109w&\_acm=get\_comm\_file&\_ ms=61479&c=131449&salt=6270021082060555853

https://editor.copernicus.org/index.php/esd-2017-79-AC2.pdf? mdl=msover md& jrl=430& lcm=oc108lcm109w& acm=get comm file& ms=61479&c=131648&salt=6370064861278646899

Here we present the **List of changes** that we have done in the main manuscript, as well as the additional information that was included to the *Supporting Information*

- To take into account the comment of the first reviewer about the appropriateness of "comparing the data between two fixed time period," in the Supporting information we include a new section where we compare relative variations computed in three different time intervals. We show that maps with very similar spatial structures are found, confirming the robustness of our findings.
- To take into account the comments of the two reviewers, we have re-written several paragraphs of the introduction.
- To take into account the comments of both reviewers, we have added to the significance analysis presented in the Supporting information, a percentile-based significance test.
- To take into account the comment of the first reviewer, we have modified the title and the text, removing the terms "interdecadal changes" and "large-scale patterns".
- To take into account the comment of the first reviewer, we have revised the conclusions and re-worded the sentence "confirm this migration in the XX century"
- As suggested by the second reviewer, we include in the Supporting information a new section with an overview of the Hilbert Transform and two figures displaying simple examples. Here we also clarify the technical comments (detrending: we remove the linear trend, we explain how the unwrapping is done and explain how the 5% is chosen, and what we mean by "exactly reconstructs").
- As suggested by the second reviewer, in the revised text we included a sentence about the quality of the reanalysis in the blue spot in the Arctic.
- We have re-written the sentence about "fast oscillations".

[revised manuscript text omitted]

---

## Author Response (AR2)

Terrassa, Barcelona, Spain, April 3, 2018

Manuscript number: ESD-2017-79

Title: Quantifying changes in surface air temperature dynamics over several decades

Authors: D. Zapalla, M. Barreiro and C. Masoller

We have answered the comments of the two referees:

Taking into account the Editor's request, we have revised the main manuscript (in page 3, red color) and the *Supporting Information* (last sentences in the section "Overview of Hilbert transform") to clarify that the reconstruction of the time series x(t) is, within the limits of numerical precision, exact; however, the values of amplitude, phase and frequency, taken separately, near the extremes of the series deviate from the true values.

Sincerely yours

Cristina Masoller
Universitat Politècnica de Catalunya
Cristina.masoller@upc.edu